# Pro-Inflammatory Priming of Umbilical Cord Mesenchymal Stromal Cells Alters the Protein Cargo of Their Extracellular Vesicles

**DOI:** 10.3390/cells9030726

**Published:** 2020-03-16

**Authors:** Mairead Hyland, Claire Mennan, Emma Wilson, Aled Clayton, Oksana Kehoe

**Affiliations:** 1School of Medicine, Keele University at the RJAH Orthopaedic Hospital, Oswestry SY10 7AG, UK; m.hyland@keele.ac.uk; 2School of Pharmacy and Bioengineering at the RJAH Orthopaedic Hospital, Oswestry SY10 7AG, UK; Claire.mennan@nhs.net; 3Chester Medical School, University of Chester, Chester CH2 1BR, UK; e.wilson@chester.ac.uk; 4School of Medicine, Cardiff CF14 4XN, UK; ClaytonA@cardiff.ac.uk

**Keywords:** umbilical cord mesenchymal stromal cell, extracellular vesicles, protein expression, pro-inflammatory priming, hypoxia, culture conditions, immunomodulation

## Abstract

Umbilical cord mesenchymal stromal cells (UCMSCs) have shown an ability to modulate the immune system through the secretion of paracrine mediators, such as extracellular vesicles (EVs). However, the culture conditions that UCMSCs are grown in can alter their secretome and thereby affect their immunomodulatory potential. UCMSCs are commonly cultured at 21% O_2_ in vitro, but recent research is exploring their growth at lower oxygen conditions to emulate circulating oxygen levels in vivo. Additionally, a pro-inflammatory culture environment is known to enhance UCMSC anti-inflammatory potential. Therefore, this paper examined EVs from UCMSCs grown in normal oxygen (21% O_2_), low oxygen (5% O_2_) and pro-inflammatory conditions to see the impact of culture conditions on the EV profile. EVs were isolated from UCMSC conditioned media and characterised based on size, morphology and surface marker expression. EV protein cargo was analysed using a proximity-based extension assay. Results showed that EVs had a similar size and morphology. Differences were found in EV protein cargo, with pro-inflammatory primed EVs showing an increase in proteins associated with chemotaxis and angiogenesis. This showed that the UCMSC culture environment could alter the EV protein profile and might have downstream implications for their functions in immunomodulation.

## 1. Introduction

Recent research has shown that mesenchymal stromal cells (MSCs) can reduce inflammation through the secretion of multiple soluble paracrine factors, such as transforming growth factor-beta 1 (TGFβ1), hepatocyte growth factor (HGF), interleukin-10 (IL-10) [1] and indoleamine 2,3-dioxygenase (IDO) [2]. Other paracrine factors secreted by MSCs include extracellular vesicles (EVs). EVs are membrane-bounded vesicles secreted by cells with a size ranging from 30 nm to over 1 micron, and they contain proteins, mRNAs, miRNAs and lipids as part of their cargo [3]. This cargo can be released into target cells and thereby change the function of the cell [4]. The cargo is often representative of their cell of origin [3]; this is thought to be why EVs derived from umbilical cord mesenchymal stromal cells (UCMSCs) have shown similar immunosuppressive effects to the UCMSCs themselves [2,5]. Indeed it has been shown in vitro that UCMSC-EVs can inhibit T-cell proliferation [6], and also in animal studies, where UCMSC-EVs are found to ameliorate hyperglycaemia in rats with type 2 diabetes mellitus [7], reduce neuroinflammation in a rat brain injury model [8] and improve acute respiratory distress symptoms in rats [9]. The evidence that UCMSCs and their EVs can suppress immune responses is appealing; however, in order to harness the full immunomodulatory properties of UCMSCs and their EVs, the best conditions for growing these cells must be established.

The MSCs secretome is variable, and the conditions MSCs are exposed to can change their functional properties [10]. For example, MSC-EVs from cells primed with pro-inflammatory cytokines (Tumor necrosis alpha (TNF-α), Interleukin 1 beta (IL-1β) and Interferon gamma (IFN-γ) have shown enhanced immunosuppressive properties against T-cells [11]. Likewise, MSC-EVs derived from cells grown in hypoxia (5% O_2_) have also shown an enhanced ability to polarise M1 macrophages into M2 macrophages in mice compared to the MSC-EVs derived from normal oxygen conditions [12]. Although there is evidence of an increased therapeutic effect of priming MSCs with low oxygen and pro-inflammatory cytokines, there is variability in methods used across the literature and no clear consensus on the best conditions to culture MSCs in order to elicit their optimal therapeutic effects. This paper aimed to characterise EV populations from different conditions (normoxia, hypoxia ± proinflammatory conditions) to see if the EV phenotype and cargo varies. It would also help provide a clearer picture of the influence of different culture methods for UCMSCs so that researchers could tailor their experiments for optimal therapeutic potency.

This was the first study comparing UCMSC-EVs from cells grown in these specific conditions. Our results demonstrated that the general biophysics of EVs (morphology, size) was similar across all conditions, but the protein cargo was altered in the pro-inflammatory primed EVs. This information on EV protein cargo is a step towards understanding the biological characteristics of UCMSC-EVs.

## 2. Materials and Methods

### 2.1. UCMSC Culture Conditions

Umbilical cords (UCs) were collected with ethical approval (National Research Ethics Service; 10/H10130/62), and research was carried out according to the Helsinki Declaration. MSCs were isolated from the whole umbilical cord, as previously described [13]. UCMSCs were first cultured expanded in a Quantum^®^ Cell Expansion System (*n* = 4) (Terumo BCT, Surrey, UK) to passage 1 and then grown on plastic thereafter. UCMSCs were grown under normoxic conditions (21% O_2_) and hypoxic conditions (5% O_2_). UCMSCs were fed every 2–3 days with DMEM F12, 10% fetal bovine serum (FBS), 1% penicillin-streptomycin (P/S) (Life Technologies, Warrington, UK). The oxygen content of the DMEM F12 was lowered to approximately 5% using the HypoxyCOOL™ media conditioning system (Baker Ruskinn, Bridgend, UK) for 3 h before adding to cells into the InvivO_2_ hypoxic workstation (Baker Ruskinn, Bridgend, UK). At 80% confluence, cells were washed with PBS, and DMEM F12 with 10% FBS (EV-depleted) was added for 48 h. Population doubling times (PDT) were calculated using the formula DT = T ln2/ln(Xe/Xb), where T is the incubation time, Xb is the cell number at the beginning of the incubation time, and Xe is the cell number at the end of the incubation time.

### 2.2. Pro-Inflammatory Priming of UCMSCs

In one experimental setting, UCMSCs were stimulated with pro-inflammatory cytokines (hereafter referred to as ‘primed’) for 48 h when they reached 80% confluence. They were treated with an inflammatory cocktail containing 5 ng/mL TNF-α, 2.5n g/mL IFN-γ and 2.5 ng/mL IL-1β (Peprotech, London, UK) [11,14,15]. Figure 1 outlines the experimental plan and culture conditions of UCMSCs.

### 2.3. Depletion of EVs from FBS

To deplete FBS of EVs, FBS was loaded into 25PC polycarbonate thick-walled centrifuge tubes (Koki Holdings Co. Tokyo, Japan) and ultracentrifuged at 120,000× *g* for 18 h at 4 °C [16] using a Hitachi Himac Micro Ultracentrifuge CS150NX (Koki Holdings Co., Tokyo, Japan). The FBS supernatant was subject to 0.2 µm filtration followed by 0.1 µm filtration.

### 2.4. UCMSC Surface Marker Characterisation

UCMSCs (*n* = 4) were characterized using flow cytometry to confirm that cells were of a mesenchymal origin. UCMSCs were harvested at passage 3–5, centrifuged at 500× *g* for 5 min and resuspended in PBS with 2% bovine serum albumin (BSA). Cells were incubated with Human BD Fc Block™ (BD Biosciences, Wokingham, UK) for 1 h; cell suspension was then centrifuged at 500× *g* for 5 min in 2% BSA, and the supernatant was removed. Cells were resuspended in 2% BSA and conjugated monoclonal antibodies against human surface antigens. The cells with antibodies were incubated in the dark at 4 °C for 30 min. The monoclonal antibodies are displayed in Appendix A. Control samples were stained with IgG controls. Flow cytometry was performed on a FACSCanto II (BD Biosciences, Wokingham, UK), and data were analysed using FlowJo^®^ software (FlowJo LLC, Ashland, OR, USA).

### 2.5. Isolation of EVs

EV isolation was carried out on UCMSC conditioned media, previously stored at −80 °C and thawed on the day of isolation. To isolate EVs, the conditioned medium underwent differential ultracentrifugation on a 30% sucrose cushion [17] using an L8-M Ultracentrifuge (Beckman Coulter, High Wycombe, UK). The conditioned media was first centrifuged at 2000× *g* for 20 min to remove cell debris. The supernatant was passed through a 0.22 µm filter (Starstedt, Leicester, UK), loaded onto a 30% sucrose cushion and centrifuged at 100,000× *g* for 1 hr 45 min on an SW28Ti rotor (Beckman Coulter, High Wycombe, UK). The EV suspension was subject to final ultracentrifugation at 100,000× *g* for 60 min on a Type 70.1 Ti fixed angle rotor to pellet pure EVs. All ultracentrifugations were performed at 4 °C, and EV pellets were stored at −80 °C.

### 2.6. Analysis of EV Morphology

Nanoparticle tracking analysis (NTA) of EV isolates and conditioned media was performed to determine particle size distributions. Isolated EV samples and conditioned media were diluted 1:50 in PBS and analysed by Nanosight NS300 system (Malvern Instruments Ltd., Amesbury, UK). Specimens were added under the control of a syringe pump, under fluid flow with speed set to 50. Videos of 60 s were taken. Videos were analysed with the NTA software (version NTA2.3), and the merged triplicate measures were represented by the presented histograms. 

Images of EVs were obtained using transmission electron microscopy (TEM). Briefly, isolated EV pellets were fixed in 50 μL of 4% paraformaldehyde (Sigma-Aldrich, Poole, UK) and dropped onto copper grids coated in 0.5% formvar in chloroform (G2002, Agar Scientific, Stanstead, UK) and left for 1 min. The grid was washed on a droplet of distilled H_2_O for 1 min and then dropped into uranyl acetate (2% in 70% ethanol) (Bio-Rad, Watford, UK) for 1 min. Finally, the grid was placed into 70% ethanol for 1 min and then left to air-dry for 1 h. A JOEL-JEM 1230 Transmission Electron Microscope (SIS systems, Birmingham, UK) was used to visualise EVs at a voltage of 200–300 kV, and images were captured using a MegaView v3 Soft Imaging System (SIS systems, Birmingham, UK). Images were analysed on GIMP v2.10.12 (www.gimp.org).

### 2.7. EV Surface Marker Expression

Characterisation of common EV surface markers CD9 and CD81 was carried out using a europium-based immunoassay. EVs were added to high-binding enzyme-linked immunosorbent assay (ELISA) strips (756071; Greiner Bio-One Ltd., Stonehouse, UK) at a concentration of 10 μg/mL and left overnight. The plate was washed and blocked for 1h 30 min using 1% BSA (DY995; R&D Systems, Abingdon, UK). One microgram per millilitre of primary antibodies against proteins, including CD9 (MAB1880, R&D Systems, Abingdon, UK) and CD81 (MCA1847, Biorad, Oxford, UK), was added to the wells and 1 μg/mL of mouse IgG2a (#14-4724-85, e-Bioscience, Waltham, MA, USA) was used as an isotype control. After washing, goat anti-mouse IgG-biotinylated antibody (NEF823001EA; Perkin Elmer, Coventry, UK), diluted 1:2500, was added for 45 min followed by europium conjugated streptavidin (1:1000; 1244-360; PerkinElmer, Coventry, UK) in red assay buffer (42-02; Kaivogen Oy, Turku, Finland) for 20 min. After the final washes, europium fluorescent intensifier (42-04; Kaivogen Oy, Turku, Finland) was added, and time-resolved fluorescence was performed on a PHERAstar FS multi-mode plate reader (BMG Labtech, Aylesbury, UK).

Further characterization of 37 surface antigens (Appendix A) on EVs was performed using the MACSPlex Exosome Kit, human (Miltenyi Biotec, Woking, UK), according to the manufacturer’s instructions. EV protein was normalised to 20 µg before adding to capture beads and antibodies.

### 2.8. EV Internal Marker Expression

Protein concentration was quantified using a Pierce BCA Protein Assay Kit (Thermo Fisher Scientific, Waltham, MA, USA), according to manufacturer’s instructions. Ten micrograms of EVs and cells were lysed with 2× Laemlli buffer with β-mercaptoethanol and denatured by heating at 70 °C for 5 min. Samples were electrophoresed on a 4%–12% TGX stain-free gel (Bio-Rad, Irvine, CA, USA) at 200 Volts for 30–40 min. Samples were then blotted onto a nitrocellulose membrane using the wet transfer method overnight at 100 volts. After blotting, membranes were blocked for 2 h in 5% semi-skimmed milk in Tris-buffered saline with Tween (TTBS) followed by incubation in primary antibodies Alix (1:200), Hsp70 (1:200), GM130 (1:200) (Santa Cruz Biotechnology, Dallas, TX, USA) for 2 h at RT. After three five-minute washes, the membrane was probed with a goat anti-mouse IgG-HRP conjugated secondary antibody (1:1000 in TTBS, Life Technologies Limited, Paisley, UK). The HRP-conjugated secondary antibody was added for 1 h, and the wash step was repeated. SuperSignal West Femto Chemiluminescent Substrate (Thermo Fisher Scientific, Waltham, MA, USA) was added to the membrane and imaged with ChemiDoc™ Touch Imaging System (Bio-Rad, Hercules, CA, USA) using Image Lab v.6.0.1 software (Bio-Rad, Hercules, CA, USA).

### 2.9. Proteomic Profiling of EVs

The protein profile of EVs was examined using a proximity extension assay (PEA) to see if cell culture conditions altered selected proteins in the EV cargo. This assay analysed 92 inflammatory associated proteins to picogram level. A comparison was made between four different conditions: 1) normoxic vs. hypoxic, 2) normoxic vs. normoxic/primed, 3) hypoxic vs. hypoxic/primed, 4) normoxic/primed vs. hypoxic/primed. EVs from four donors were included in each group. Isolated EVs were lysed with RIPA buffer (Merck, Poole, UK) and 1x protein inhibitor (Roche, Mannheim, Germany). Protein concentration was analysed using a BCA protein kit (Thermo Fisher, Paisley, UK). Samples were normalised to a protein concentration of 0.5 mg/mL and analysed by a sensitive proximity ligation assay. The Proseek^®^ Multiplex Inflammation Panel was used (Olink Bioscience, Uppsala, Sweden) to detect the presence of 92 proteins (Appendix A). The assay uses two oligonucleotide-conjugated antibodies that bind to protein targets. Upon binding to the protein epitope, the paired oligonucleotide sequences are amplified through a quantitative real-time PCR (qRT-PCR) reaction. Data is then generated using normalized protein expression (NPX) values on a log2 scale whereby a higher NPX correlates with higher protein expression. More detailed information is available on Olink’s website (www.olink.com). Proteins containing NPX values >50% below the assay’s limit of detection (LOD) were excluded from the analysis. For the data analysis, samples <LOD were substituted with LOD/$\sqrt{2}$.

### 2.10. Statistical Analysis

All statistical analysis was performed on GraphPad Prism v8 (GraphPad Software, San Diego, CA, USA). All analysis was carried out using paired students *t*-tests. Differential protein expression from the PEA assay was analysed using multiple *t*-tests with a false discovery rate (FDR) of 5% to control for multiple comparisons. FDR-adjusted *p*-values (referred to as *q*-values) <0.05 were considered significant.

## 3. Results

### 3.1. Stromal Cells Isolated from Umbilical Cords Exhibited an MSC Phenotype and Showed Similar Growth Kinetics in Normoxic and Hypoxic Conditions

As part of the International Society for Cellular Therapy criteria for characterising MSCs, they must be ≥95% positive for the surface antigens CD105, CD90, CD73 and ≤2% negative for the surface antigens CD14, CD19, CD34, CD45 and HLA-DR [18]. Our results showed that the UCMSCs matched these criteria (Appendix A). UCMSCs were previously characterised using tri-lineage differentiation [13] and displayed positive differentiation down osteogenic and adipogenic lineages and a low degree of chondrogenic differentiation. The average PDT for UCMSCs grown in normoxic conditions, from P3-8, was 2.82 days ± 0.2 compared to 2.52 ± 0.38 days for cells grown in hypoxic conditions. There were no statistical differences between the conditions (Figure 2).

### 3.2. EVs Displayed a Similar Size, Morphological Appearance and Surface Marker Expression across All Conditions

The EVs isolated from different culture conditions did not show any size differences when analysed using NTA. The average size of the purified UCMSC-EVs was 167 nm ± 7 nm, and the average size of particles in the unpurified conditioned media was 171 nm ± 35 nm (Figure 3A). There was a more diverse size range of particles from the conditioned media. After the EV isolation process, there was a more uniform size of particles showing that EVs had been purified during the process (Figure 3B). Pre-centrifugation of FBS was successful in reducing the protein concentration of EVs by 78% compared to EVs isolated from untreated FBS (data not shown). TEM analysis showed that isolated EVs from UCMSCs had a similar round morphological appearance with a visible phospholipid bilayer (Figure 3C).

The tetraspanins CD9, CD81 and CD63 are common EV surface markers included in the International Society of Extracellular Vesicle (ISEV) guidelines for characterising EVs [19]. These surface markers were identified in all EV samples. The amount of CD9 and CD81 was statistically higher in hypoxic compared to hypoxic/primed EVs (*p* < 0.05) from the immunoassay (Figure 4). There were no differences found amongst other conditions. Results from the Macsplex exosome detection kit supported the finding that EVs contained surface markers CD9 and CD81. The EVs also contained the tetraspanin marker CD63, MSC marker CD105 and markers associated with cell adhesion and migration: CD29, CD44, CD49e, but these levels didn’t change statistically between conditions (Appendix A).

Immunoblotting analysis was carried out on UCMSCs and their corresponding EVs to detect for the presence of cytosolic proteins in EVs; Alix and Hsp70 in normoxic, hypoxic and normoxic/primed conditions. Positive expression of Alix (96 kDa) and Hsp70 (70 kDa) was identified in UCMSCs and UCMSC-EVs. GM130, a negative marker for EVs, was found only in cell samples (Appendix A).

### 3.3. Pro-Inflammatory Priming Altered EV Protein Cargo

In order to explore possible changes in the protein-repertoire of EV imposed by varied culture conditions, we performed a protein profiling experiment using a highly sensitive proximity-ligation array. Results from this array found no statistical differences in EV protein cargo between group 1 normoxic vs. hypoxic and group 4 normoxic/primed vs. hypoxic/primed group. We observed statistical differences between EVs from cells with or without pro-inflammatory priming (Group 2 and 3) with the pro-inflammatory groups showing a higher abundance in the expression of chemotactic and angiogenic proteins. Of the 92 inflammatory-related proteins in the assay, 52 were detected in the EV samples (Appendix A) with 16 proteins showing statistical differences in the normoxic vs. normoxic/primed group and 21 proteins showing statistical differences in the hypoxic vs. hypoxic/primed group. All samples passed internal and external plate quality controls.

Table 1 depicts the difference in Log_2_(fold change) in protein expression between the EVs from primed/non-primed groups. INF-γ showed the highest fold change in both groups. There was also a considerable elevation in the proteins CSF-1, MCP2, MCP4 and CCL3, which showed at least a 10-fold higher change in primed compared to non-primed conditions. The proteins TGF-α and IL-13 were only detected in primed conditions. In total, there were 15 proteins with increased expression in the normoxic vs. normoxic/primed group (Figure 5A) and 19 proteins with increased expression in the hypoxic vs. hypoxic/primed group (Figure 5B), of which 12 proteins overlapped with both groups. The normoxic/primed EVs had a decreased amount of one protein (CXCL10) in comparison to normoxic EVs. Additionally, hypoxic/primed EVs had a decreased amount of two proteins (CXCL10 and CXCL6) in comparison to the hypoxic EVs.

Other noteworthy proteins that were detected in the EV samples that are known to have anti-inflammatory properties include TGF-β1, FGF5, FGF21, PDL1, HGF, STAMBP; however, these were not differentially expressed between conditions. Similarly, proteins with pro-inflammatory properties, such as IL-8, MCP1, CXCL11, CXCL9, IL-1a, CXCL1, MMP1, IL15RA, IL12b, CD40, were also found in EVs but not differentially expressed between conditions. Other potentially anti-inflammatory proteins, such as IL-4 and IL-10, were not detected in the EVs.

## 4. Discussion

Culture conditions can be tailored in the lab to alter the MSC secretome [10], but it remains to be answered which culture conditions generate EVs containing an enriched anti-inflammatory cargo for treatment of inflammatory conditions. This study explored the EV profile from USMSCs grown in four different cell culture conditions with an aim towards characterising and comparing the EVs cargo from each condition.

This study found that EV morphology and size did not differ between conditions. This is not unexpected as previous research has also found similar results, showing no statistical differences in size, shape and electrodensity between EVs from normoxic and hypoxic conditions [12], but a study by Varkouhi et al. found that IFN-γ primed EVs were approximately 1.5 times larger than their non-primed counterparts; however, this study used a higher concentration of IFN-γ for a shorter time than the current study [9]. In all, it remains unclear if subtle changes in EV size has a major bearing on their downstream paracrine functions.

Production of EVs positive for surface markers (CD9, CD63, CD81) and internal markers (Alix, Hsp70) was identified, and this data, accompanied by EV size and morphological analysis, showed that the isolated EV population was in accordance with the ISEV guidelines [19]. There was also little difference found in surface marker expression except for an increase of CD9 and CD81 in hypoxic EVs compared to hypoxic/primed EVs. This elevated level of CD9/CD81 could be explained by hypoxic/primed cells producing and secreting more of the CD9/CD81 subpopulation and thereby altering the heterogeneity of the EV under this condition; however, the larger sample size is needed as there is a high variability amongst donors. Differences in tetraspanin surface markers have also been found in EVs derived from different cell culture conditions. Lo Sicco et al. found that CD63, another EV surface marker, was higher in hypoxic EVs compared to normoxic EVs, showing that EV surface markers could be altered by their environment [12]. But this marker was not probed for in the europium immunoassay, and no differences were found between conditions in the levels of CD63 from the Macsplex Exosome Detection kit.

The main finding here was that the EV protein cargo from UCMSCs, primed with pro-inflammatory cytokines, was altered compared to the non-primed groups. In total, the EVs from primed conditions showed an increased pro-inflammatory proteomic profile. The increased production of TNF and IFN-γ in primed EVs is not surprising as the cells were primed with these pro-inflammatory cytokines, which could have been absorbed by cells and packaged into EVs. TNF and IFN-γ could also have been bound to the membrane of EVs, which has been found in previous studies [20]. After this, the CCL3 protein had the highest fold change in both normoxic/primed and hypoxic/primed groups. In addition, MCP2, MCP4 and CSF-1 also showed at least a 10-fold change in the primed conditions. Overall, primed EVs showed a general trend towards a more chemotactic and pro-inflammatory profile, with the exception of CXCL10 and CXCL6. This suggested that the priming of UCMSCs with a cytokine cocktail was polarising them towards a similar phenotype.

MSCs are a heterogeneous group of cells, and they express both pro-inflammatory and anti-inflammatory proteins when primed [14]. Therefore, it was not unusual to see EVs from primed UCMSCs displaying pro-inflammatory proteins. Other groups have shown that bone marrow MSCs, stimulated by TNF-α and IFN-γ, secreted a greater number of pro-inflammatory cytokines CCL5, CXCL9, CXCL10, CXCL11, IL12 and IL15 compared to the unstimulated group [14]. However, research also shows that this increase in pro-inflammatory cytokines correlates with an increase in the level of anti-inflammatory molecules (IDO, PD-L1, IL-4, IL-10 and HLA-G) [14]. This finding deviates from the current study as PD-L1 levels did not statistically differ between conditions; IL4 and IL10 were not detected; IDO and HLA-G were not probed for. The cargo of USMSC-EVs in this study contained many chemokines (CXCL5, CCL3, CCL4, CCL11, CCL20), which might attract immune cells to the site of injury. However, research has strongly shown that MSCs primed with pro-inflammatory cytokines (IFN-γ, TNF-α, IL-1β) show enhanced immunosuppressive abilities [14,21,22,23]. Specifically, research using EVs secreted from pro-inflammatory primed MSCs has shown positive immunosuppressive effects by polarising pro-inflammatory immune cells into anti-inflammatory immune cells in vitro [24,25,26,27,28,29].

This study found no significant difference in growth kinetics of protein cargo between UCMSCs grown in normoxia and hypoxia, and research was divided on whether hypoxia has a stimulatory [30] or inhibitory [31,32] effect on MSC proliferation; albeit studies varied between the use of 1%-5% O_2_. Some groups have advocated for the use of hypoxia when growing cells in culture as it more closely mimics the oxygen levels that MSCs are exposed to in vivo [22], with neonatal-derived tissues rarely exceeding 5% in vivo [23]. The lack of change in growth may be because the cells were first isolated under normal oxygen (21%) conditions, and exposure to low oxygen (5%) at an earlier stage may be an influencing factor in cell proliferation. Despite UCMSCs having similar PDTs between conditions, hypoxia is still worth exploring as animal studies have shown that UCMSCs, grown in hypoxia, have a higher angiogenic potential in the treatment of mouse hindlimb ischaemia [33,34] and rat spinal cord injury in comparison to UCMSCs grown in normoxia [35]. Indeed, the pro-angiogenic protein VEGF-A [36,37,38], was expressed in all conditions but upregulated in hypoxic/primed EV samples. VEGF-A is a well-documented hypoxic-induced response [39], there might be other hypoxic related changes in the EVs that weren’t detected in this array; an array covering a broader repertoire of proteins might have identified more hypoxic-mediated changes.

When trying to decipher if primed EVs have an immunosuppressive potential, two proteins came to attention, namely, leukaemia inhibitory factor (LIF), which was upregulated in hypoxic/primed EVs, and MMP10 in normoxic/primed EVs. Both these proteins have anti-inflammatory potential. LIF has been found to suppress T-helper 17 cells in an experimental autoimmune encephalomyelitis mouse model [40] and promote Treg proliferation in an MSC-mixed lymphocyte co-culture [41]. The identification of LIF in UCMSC-EVs may also cause similar immunosuppressive effects. With regards to MMP10, the MMP family of proteins are widely secreted by MSCs [42], and they play a role in breaking down the extracellular matrix [43], but recent research has shown that they have immunomodulatory effects through cleaving CC chemokines, such as CCL2 (MCP1). This process inhibits the function of CCL2, and it cannot activate an immune response but instead suppresses T-cell chemotaxis [36,42,44,45]. As MMPs have a role in cleaving CC chemokines, they may have anti-inflammatory functions in vivo. This was interesting for the EVs in this study as they contained MMP1 and MMP10, which might target CC chemokines, such as MCP2, MCP3 and MCP4, also found in EV samples. Other anti-inflammatory proteins, HGF and FGF21, were expressed in EVs, but their level didn’t vary between samples. HGF protein has been found to decrease IL-6, increase IL-10 expression [46] and polarise Th-1 cells into a Th-2 phenotype [47]. FGF21 also has anti-inflammatory functions through suppressing TNF-α, IL-1β, IL-6, IFN-γ and increasing IL-10 macrophages [48]. It could be possible that UCMSCs in this study also secreted more anti-inflammatory proteins in the form of soluble factors that were just not being packaged into EVs.

Overall, the pro-inflammatory-primed EVs in this study produced pro-inflammatory proteins, which might act to trigger inflammation in vivo. There was no change in the production of anti-inflammatory proteins—TGF-β1, PDL1 and HGF—between conditions. There are limitations in this study, which should be considered when interpreting the data. Firstly, our study had a sample size of four, and as MSC-EVs are heterogeneous, we are working on obtaining a larger sample size to identify further differences between the conditions. Additionally, this study looked solely at the basic characterisation of UCMSC-EVs; and functional studies would be required to see if this EV population alters T-cell polarisation and proliferation, in order to determine their therapeutic potential. These functional studies are especially needed as the PEA assay only analysed proteins associated with inflammation, and other potential therapeutic proteins may not have been probed in the assay but may play an important immunosuppressive role.

## 5. Conclusions

This research represented the first of its kind in characterising EVs from UCMSCs grown in four different culture conditions. It aimed to identify how culture conditions could affect MSC-EV cargo. This type of research is needed to garner the best possible outcomes for future UCMSC-EV therapy. There were no significant changes seen in EVs grown in different conditions with regard to morphology, size and internal EV markers. One difference was found in the expression of EV surface markers, where hypoxic EVs contained more CD9 and CD81 compared to the hypoxic/primed EVs; however, it is not yet clear if these differences would translate to changes in EV functioning. The proximity-based extension assay did not show any differences in the expression of inflammatory proteins between the normoxic and hypoxic conditions. The main changes in EV protein cargo occurred between primed and non-primed EVs, which saw an increase in proteins associated with chemotaxis and angiogenesis. However, there still might be other differences between these groups, which were not probed for in the 92-protein inflammation panel. This study only looked at one piece of the puzzle, and a more comprehensive study into both the transcriptome and proteome of EVs is needed to discover how they may have immunomodulatory functions. Some of the proteins expressed in EVs are pleiotropic in nature, and functional studies are, therefore, required to analyse their effect in an activated immune environment. Results from these functional tests would help to evaluate the immunomodulatory ability of UCMSC-EVs.

## Figures and Tables

**Figure 1 cells-09-00726-f001:**
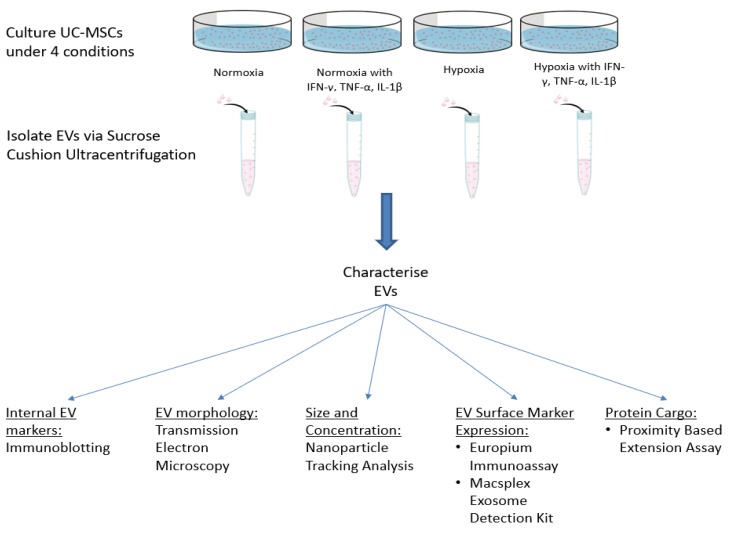
Schematic of the study plan, including culture conditions of UCMSCs and EV characterisation experiments.

**Figure 2 cells-09-00726-f002:**
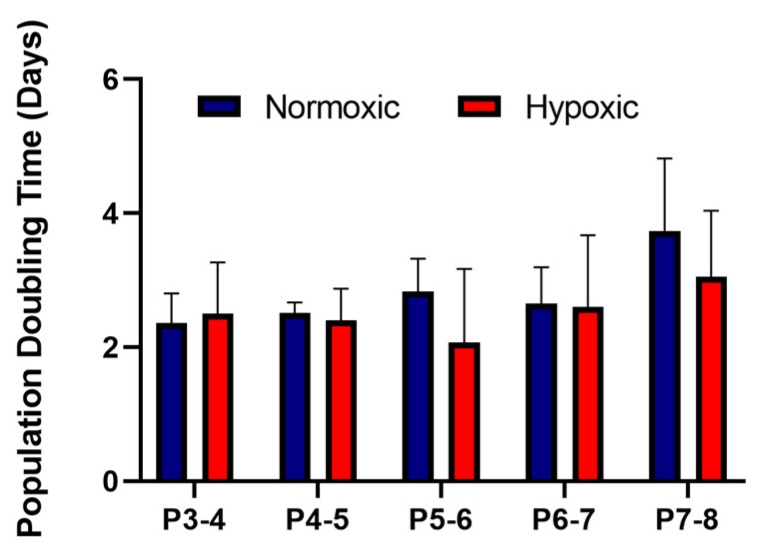
Average population doubling times for umbilical cord mesenchymal stromal cells (UCMSCs) (*n* = 4) grown in normoxia (with 21% O_2_) and hypoxia (with 5% O_2_). Cell counts were performed at each passage from P3-8. Error bars indicate mean ± SD.

**Figure 3 cells-09-00726-f003:**
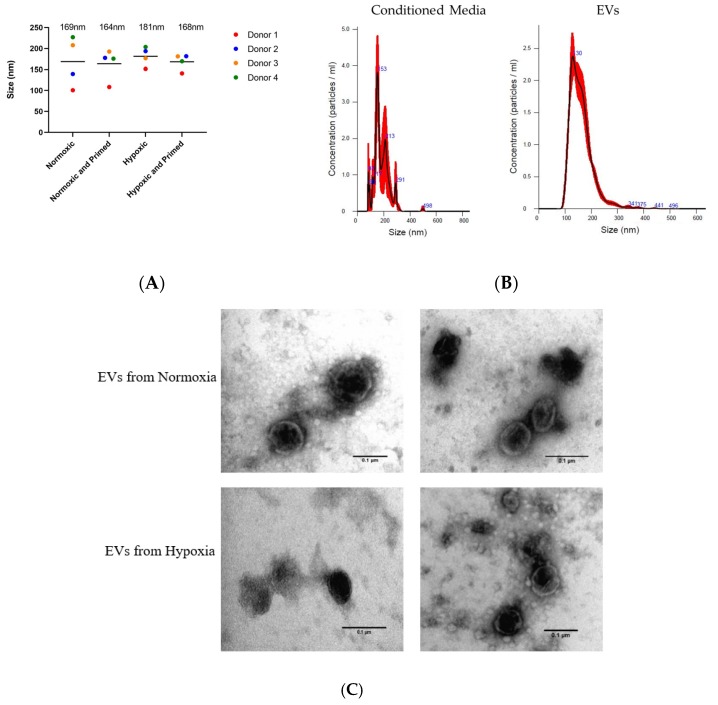
(**A**) Size of particles in the conditioned media from the four donor’s samples in different conditions based on NTA data. The average size of particles for each condition is displayed above plots. (**B**) Size profile of particles from conditioned media and particles from isolated extracellular vesicles (EVs). (**C**) TEM images of EVs in normoxia (first two) and hypoxia (last two). Scale bars = 0.1 µm.

**Figure 4 cells-09-00726-f004:**
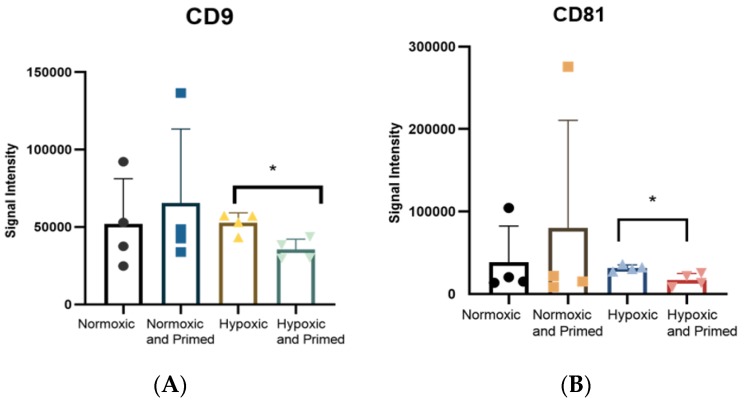
Histogram showing the expression of CD9 (**A**) and CD81 (**B**), detected using a europium-based immunoassay. Statistical differences were found between CD9 and CD81 in the hypoxic EVs compared to the hypoxic/primed EVs. Error bars indicate mean ±SD. * *p* < 0.05.

**Figure 5 cells-09-00726-f005:**
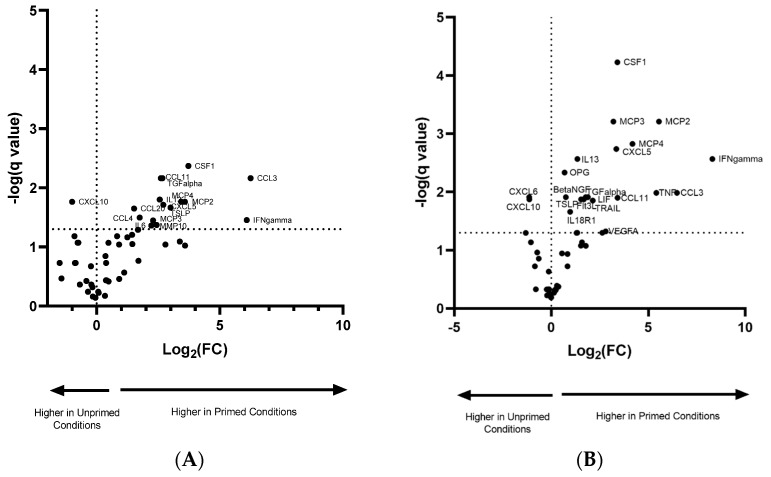
Volcano plot showing the normalized protein expression (NPX) difference in protein expression in EVs from normoxic and normoxic/primed conditions (**A**) and hypoxic and hypoxic/primed conditions (**B**). There was increased expression in 15 proteins from the normoxic/primed condition and a decreased expression in 1 protein (CXCL10). There was increased expression in 19 proteins from the hypoxic/primed condition and a decreased expression in 2 proteins (CXCL6, CXCL10). Values above the horizontal dotted line on the x-axis are statistically significant. Values to the left of the vertical dotted line on the y-axis have a decrease in NPX (Log_2_ (FC)); values to the right of this line have an increase in NPX (Log_2_ (FC)).

**Table 1 cells-09-00726-t001:** Correlations between protein expression levels from primed and non-primed conditions from cells grown in normoxia (left) and hypoxia (right). All proteins listed show statistically significant differences with a q-value <0.05. Data is presented with normalized protein expression (NPX) difference, *p*-values and adjusted *p*-values (q-values) using a false discovery rate method of 5%.

Difference in Normoxic/Primed vs. Normoxic				Difference in Hypoxic/Primed vs. Hypoxic			
Protein	Difference	*p*-Value	*q*-Value	Protein	Difference	*p*-Value	*q*-Value
CSF1	3.72	0.0001	0.0043	CSF1	3.41	0.000002	0.0001
TGF-α	2.61	0.0007	0.0069	MCP3	3.20	0.00004	0.0006
CCL11	2.67	0.0005	0.0069	MCP2	5.55	0.0001	0.0006
CCL3	6.25	0.0007	0.0069	MCP4	4.19	0.0002	0.0015
IL13	2.56	0.0020	0.0158	CXCL5	3.35	0.0003	0.0018
MCP4	3.43	0.0034	0.0172	IL13	1.35	0.0005	0.0027
CXCL10	−0.99	0.0034	0.0172	IFN-γ	8.31	0.0006	0.0027
MCP2	3.58	0.0035	0.0172	OPG	0.69	0.0011	0.0046
CXCL5	2.71	0.0044	0.0193	TNF	5.42	0.0030	0.0103
TSLP	3.00	0.0054	0.0216	CCL3	6.48	0.0031	0.0103
CCL20	1.52	0.0062	0.0224	TGF-α	1.89	0.0045	0.0122
CCL4	1.75	0.0096	0.0318	Beta-NGF	0.75	0.0049	0.0122
IFN-γ	6.09	0.0115	0.0353	CXCL6	−1.14	0.0051	0.0122
MCP3	2.29	0.0125	0.0356	LIF	1.78	0.0049	0.0122
MMP10	2.44	0.0159	0.0424	CCL11	3.42	0.0056	0.0125
IL6	2.22	0.0174	0.0434	TSLP	1.68	0.0067	0.0133
				Flt3L	1.54	0.0065	0.0133
				CXCL10	−1.13	0.0072	0.0134
				TRAIL	2.14	0.0079	0.0140
				IL18R1	0.98	0.0131	0.0219
				VEGFA	2.81	0.0296	0.0473

Beta-nerve growth factor (Beta-NGF); C-C motif chemokine 3 (CCL3); C-C motif chemokine 4 (CCL4); C-C motif chemokine 20 (CCL20); C-X-C motif chemokine 5 (CXCL5); C-X-C motif chemokine 6 (CXCL6); C-X-C motif chemokine 10 (CXCL10); Eotaxin (CCL11); FMS related tyrosine kinase 3 ligand (Flt3L); Interferon gamma (IFN-γ); Interleukin 6 (IL6); Interleukin-13 (IL13); Interleukin 18 receptor 1 (IL18R1); Leukemia inhibitory factor (LIF); Macrophage colony stimulating factor 1 (CSF1); Matrix metalloproteinase 10 (MMP10); Monocyte chemotactic protein 2 (MCP2); Monocyte chemotactic protein 3 (MCP3); Monocyte chemotactic protein 4 (MCP4); Osteoprotegerin (OPG); Thymic stromal lymphopoietin (TSLP); TNF related apoptosis inducing ligand (TRAIL); Transforming growth factor alpha (TGF-α); Tumor necrosis factor (TNF); Vascular endothelial growth factor A (VEGFA).

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
