# Peer review of "Pro-Inflammatory Priming of Umbilical Cord Mesenchymal Stromal Cells Alters the Protein Cargo of Their Extracellular Vesicles"

_cells, 2020, doi:10.3390/cells9030726_

Round 1

Reviewer 1 Report

These are extremely interesting preliminary studies on the use of stem cells isolated from the umbilical cord.
The authors showed that umbilical cord stromal mesenchymal cells (UCMSC) have the ability to modulate the immune system by secreting paracrine mediators such as extracellular vesicles. The work is well prepared and provides the basis for further research that will allow to know the properties of this cell population in terms of usefulness or not for possible future use in therapies. These are extremely important basic research, when commercial companies without proper research offer for payment "stem cells" with unknown immune status. I propose to improve the readability of schemes / figures and their numbering.

Author Response

We thank Reviewer #1 for taking time to look through our paper. We are pleased to hear that you find value in this research and we have addressed the comments about the readability of schemes, figures and numbering so that it is better presented to the reader.

Comment: I propose to improve the readability of schemes / figures and their numbering.

Response: Thank you for this comment, we have made changes to a number of figures and diagrams and we have corrected their numbering. This includes Figure 1-change to clarify of diagram; Figure 2- upper error bars added and mean ± SD added; Figure 3(C)- Labels ‘normoxia’ and ‘hypoxia’ added to images; Figure 4- CD9 and CD81 order corrected. These figures are now better presented and more readable.

Reviewer 2 Report

General comment:

The authors examined EVs from UCMSCs grown in normal oxygen (21% O2), low oxygen (5% O2) and pro-inflammatory conditions to see the impact of culture conditions on the EV profile. They found that EVs had a similar size and morphology. Differences were found EV protein cargo, with pro-inflammatory primed EVs showing an increase in proteins associated with chemotaxis and angiogenesis. The authors conclude that the UCMSC culture environment can alter the EV protein profile and may have implications for their functions in immunomodulation.

General concerns:

Page 5, 3. Results: Please correct “There were no statistical differences between the conditions (Figure 1)” to “There were no statistical differences between the conditions (Figure 2).” The Figure order is wrong. Please correct the subsequent Figure 1, 2, 3, and 4 to Figure 2, 3, 4, and 5 and the description in the text. Figure 3: “(C) TEM images of EVs in normoxia (first two) and hypoxia (last two)” was not clear. Please add normoxia and hypoxia in the figure panel C. Supplementary Figure 3: Please add “normoxic, hypoxic, normoxic/primed conditions” at the top of the figure.

Author Response

We thank this reviewer for their feedback on this paper. We welcome their suggestion to improve the figures and diagrams and feel that we have addressed this point to make our paper clearer for the reader.

Comment: Page 5, 3. Results: Please correct “There were no statistical differences between the conditions (Figure 1)” to “There were no statistical differences between the conditions (Figure 2).”

Response:  Thank you for identifying this- it has been changed to ‘Figure 2’ (Line 195)

Comment: The Figure order is wrong. Please correct the subsequent Figure 1, 2, 3, and 4 to Figure 2, 3, 4, and 5 and the description in the text.

Response: Figure order corrected-thank you.

Comment: Figure 3: “(C) TEM images of EVs in normoxia (first two) and hypoxia (last two)” was not clear. Please add normoxia and hypoxia in the figure panel C. 

Response: We have added labels to the photos to make it clear which samples were normoxia and which samples were hypoxia (Line 210).

Comment: Supplementary Figure 3: Please add “normoxic, hypoxic, normoxic/primed conditions” at the top of the figure.

Response: Labels added to top of figure on the western blot (Line 38 suppl. Materials), thank you.

Reviewer 3 Report

The manuscript cells-714725 is interesting, both considering its topic as well as the results obtained by the Authors themselves. Mesenchymal stromal cells themselves is a hot topic as (https://www.ncbi.nlm.nih.gov/pubmed/31703272) as also are exosomes-extracellular vesicles, e.g. in allergy (https://www.ncbi.nlm.nih.gov/pubmed/30195377).

The problem is that the current manuscript has, at least in my view, simply not sufficient data for the full-length article, especially in the journal of this rank. In principle, one experiment has been conducted (first Figure 1, page 5) followed by several preliminary (second Figure 1, page 6, Figure 2, Figure 3) and one primary analysis, i.e. proteomics (Table 1, Figure 4). These characterizations steps are not followed by any experiments targeting the mechanistic side of the results obtained in the characterization part, which is, at least in my opinion, not enough. Another things is that the number of the samples per group seems to be rather limited making some data difficult to interpret, especially when a substantial variability is present within the group, as in Figure 2A, the “Normoxic” panel or Figure 3A and 3B, the “Normoxic” and “Normoxic and Primed” panels.

Of the other comments, presentation of the data could be improved. For example some elements of the first Figure 1, page 5 are either too small or not sharp enough. In the second Figure 1, page 6, upper error bars would suffice; in Figure 3, if SD (see further) as well. Besides, the presentation of the data always differs, including means +/- SD (second Figure 1, page 6), real data + means (? Statistically looking like means but graphically looking like medians; Figure 2A), or real data + means + SD (?; Figure 3).

Author Response

We thank reviewer #3 for their suggestions for amendments to the figures in this report. The changes made have been valuable and have improved the clarity of the manuscript.

Comment: The problem is that the current manuscript has, at least in my view, simply not sufficient data for the full-length article, especially in the journal of this rank. In principle, one experiment has been conducted (first Figure 1, page 5) followed by several preliminary (second Figure 1, page 6, Figure 2, Figure 3) and one primary analysis, i.e. proteomics (Table 1, Figure 4). These characterizations steps are not followed by any experiments targeting the mechanistic side of the results obtained in the characterization part, which is, at least in my opinion, not enough.

Response: We thank the reviewer for the helpful comments and apologise that Figure 1 is labelled incorrectly in two places- we have corrected this.  We feel that the aim of this paper was not to present a targeted mechanistic paper but to present the preliminary biologic characterisation of EVs isolated from UC-MSCs cultured in normoxia and hypoxia with the addition of pro-inflammatory stimulus.  To our knowledge this has not been published before and feel that it would be a useful addition to the field of EV research. 

The authors agree that studying the mechanistic side of these EVs would indeed be an interesting and a crucial experiment to perform in order to build on these results. Further work in our group is aiming to carry out these functional experiments on the assessment of EV phenotypes on immune modulation using lymphocytes as part of an ongoing Ph.D studentship, but has not been within the scope of this current project.

We have further highlighted in the introduction, discussion and conclusion that this paper presents data solely on the characterisation of EVs without eluding to their mechanistic properties and we hope that this will make it clearer to the readership of the journal.

Lines 56-58 (original doc) “This characterisation aims to identify a population of EVs with the best therapeutic potential to supress immune cells in the treatment of autoimmune diseases”.

Has been changed to read:

Lines 57-59 “This paper aims to characterise EV populations from different conditions (normoxia, hypoxia ± proinflammatory conditions) to see if the EV phenotype and cargo varies”.

Line 62-63 (original doc): This information on EV protein cargo is a step towards understanding how UCMSC-EVs maybe capable of suppressing immune cells.

Has been changed to read:

Line 63-64: This information on EV protein cargo is a step towards understanding the biological characteristics of UCMSC-EVs.

Line 362-363 (original doc) ‘Functional studies will be beneficial to see if this EV population alters T-cell polarisation and proliferation’

Has been changed to read:

Line 367-369 (when discussing limitations) ‘Additionally, this study looked solely at the basic characterisation of UCMSC-EVs; and functional studies will be required to see if this EV population alters T-cell polarisation and proliferation, in order to determine their therapeutic potential’

We have changed the first sentence in the conclusion

Line 368-369 (original doc): ‘This research characterised EVs from UCMSCs grown in different conditions with one aim of interpreting how UCMSCs carry out their immunomodulatory effects’

Has been changed to

(Line 374-375) ‘This research represents the first of its kind in characterising EVs from UCMSCs grown in four different culture conditions. It aimed to identify how culture conditions can affect MSC-EV cargo’, so that it is not misleading and clear to the reader.

Comment: Another things is that the number of the samples per group seems to be rather limited making some data difficult to interpret, especially when a substantial variability is present within the group, as in Figure 2A, the “Normoxic” panel or Figure 3A and 3B, the “Normoxic” and “Normoxic and Primed” panels.

Response:

Again, we thank the reviewer for pointing out the lower n number. We agree a larger sample size would be beneficial to this study and we intend for the study to be expanded on in the future. However, we believe that 4 samples represent a valuable and publishable set of data.  Indeed, the data does show variability between samples, however, we think it is generally accepted within the MSC and EV field that there is significant donor-to-donor variation between samples, and both cells and EVs are heterogeneous. The donor-to-donor variation between MSC samples has been well reported (Wilson et al., 2019). Previous work in our group has also shown similar results with the UC-MSCs and bone marrow derived (BM)-MSCs i.e. variation in cell growth, response to inflammatory stimulus and ability to differentiate in vitro. Other groups have also shown this donor variation with cell growth in UCMSCs (Han et al., 2017), growth and immunosuppression in cord blood MSCs (Amati et al., 2017) and Francois et al. (2012) showed that bone marrow MSCs had differing immunomodulatory properties between donors. This variation also extends to the EV field with Bobis-Wozowicz et al. (2017) finding donor variability in UCMSC-EV samples (n=5) in terms of surface marker, internal marker and mRNA expression.

Furthermore, 4 donors’ cells were each culture expanded in a large scale cell expansion system (Quantum®) for this work in order to generate the cell numbers required to collect EVs for basic characterisation.  Again, culturing more donors’ cells in this system was beyond the scope of this project.  However, the characterisation of EVs from Quantum derived UC-MSCs represents a further novel aspect of this study as it has not previously been published. 

One of the aims of our future work is to find MSC donors that have the most potent/best EVs for treating patients.  We have mentioned the 4 donors as a limitation in terms of EV surface marker expression (Line 295-296), but we have also added a line in the discussion to highlight this variability and the benefit of a larger sample size to see more differences (Line 364-367) -  “There are limitations in this study which should be considered when interpreting the data. Firstly, our study has a sample size of four and as MSC-EVs are heterogenous, we are working on obtaining a larger sample size to identify further differences between the conditions”.

In all, there are few published studies on EVs from UC-MSCs and none comparing the specific culture conditions detailed in this study, therefore, although the results show donor variation we think it is worthy of publication given that the limitations have now been further highlighted in the manuscript and hope that the reviewer agrees.

Wilson A, Hodgson-Garms M, Frith JE, Genever P. Multiplicity of Mesenchymal Stromal Cells: Finding the Right Route to Therapy. Front Immunol. 2019;10:1112. Published 2019 May 16. doi:10.3389/fimmu.2019.01112

Han, Z.C., Du, W.J., Han Z.B. & Liang, L. ‘New Insights into the Heterogeneity and Functional Diversity of Human Mesenchymal Stem Cells’. 1 Jan. 2017: S29 – S45

Amati, E., Sella, S., Perbellini, O., Alghisi, A., Bernardi, M., Chieregato, K., Lievore, C., Peserico, D., Rigno, M., Zilio, A., Ruggeri, M., Rodeghiero, F., & Astori, G. “Generation of mesenchymal stromal cells from cord blood: evaluation of in vitro quality parameters prior to clinical use.” Stem cell research & therapy vol. 8,1 14. 24 Jan. 2017, doi:10.1186/s13287-016-0465-2

François, M., Romieu-Mourez, R., Li, M.and Galipeau, J. Human MSC Suppression Correlates with Cytokine Induction of Indoleamine 2,3-Dioxygenase and Bystander M2 Macrophage Differentiation. Molecular Therapy, vol. 20, 1, 187–195 jan. 2012, doi: https://doi.org/10.1038/mt.2011.189

Bobis-Wozowicz S, Kmiotek K, Kania K, Karnas E, Labedz-Maslowska A, Sekula M, Kedracka-Krok S, Kolcz J, Boruczkowski D, Madeja Z, Zuba-Surma EK. Diverse impact of xeno-free conditions on biological and regenerative properties of hUC-MSCs and their extracellular vesicles. J Mol Med (Berl). 2017 Feb;95(2):205-220. doi: 10.1007/s00109-016-1471-7.

Comment: Of the other comments, presentation of the data could be improved. For example some elements of the first Figure 1, page 5 are either too small or not sharp enough.

Response: The diagram has been made sharper and font size larger so that the reader can clearly read it. (Line 179-180)

Comment: In the second Figure 1, page 6, upper error bars would suffice in Figure 3, if SD (see further) as well.

Response: Figure on page 6, line 194, changed to just include upper error bars, and Figure 3 (now changed to Figure 4) page 8 Line 225, also changed to just include upper error bars with mean ±SD.

Comment: Besides, the presentation of the data always differs, including means +/- SD (second Figure 1, page 6), real data + means (? Statistically looking like means but graphically looking like medians; Figure 2A), or real data + means + SD (?; Figure 3)

Response: Thank you for identifying this. We have included in figure legends that the data is means SD to be clear (Line 197 & 228).

Reviewer 4 Report

The manuscript "Pro-inflammatory priming of umbilical cord mesenchymal stromal cells alters the protein cargo of their extracellular vesicles" is very interesting and deals with topics of high scientific impact: defining a standard procedure for the isolation and culture of MSCs, whatever the tissue of origin, understanding which stimulation condition would be the best in order to obtain a better therapeutic effect, represents a topic of great interest for the whole scientific community that deals with defining the possible use of these cells and their secretoma.

In particular, the authors compared EV cargoes isolated from the culture media of MSCs that underwent to different stimulation protocols, to select eventually the best EVs to be exploited in the treatment of autoimmune diseases. Cytokine stimulation represent a method widely used in the literature to enhance the immunomodulatory potential of MSCs and their EVs.

I believe that the study by Hyland and colleagues is well done and the results are convincing. However, it represents only the beginning of the study since MSCs from only 4 donors were used. Therefore, it deserves to be expanded in the future, as pointed out by the authors themselves.

Major comments

The authors showed that MSC stimulation with a cocktail of TNF-a, IFN-g and IL-1b induces an enrichment of pro inflammatory factors in EVs, indicating that the priming of UCMSCs polarize them towards a pro-inflammatory phenotype. The authors well argue that this is in contrast with what has been described in the literature where cytokine priming generally leads to an increase in the immunoregulatory capacity of MSCs and their secretoma, leading to immunosuppression.

In order to clarify this discrepancy, it would be necessary to set up functional experiments in which the EVs extracted from normoxic / hypoxic primed MSCs exert an immunostimulatory action (as the identified cargo would suggest) or immunosuppressant potential (as suggested by most studies in the literature). This could be verified by administrating EVs from the different MSCs on immune cells (e.g. lymphocytes / macrophages or even microglial cells, cell lines or primary cell cultures). It should also be envisaged to carry out the experiment considering the possibility of comparing EVs extracted from culture media not subjected to freezing.

The reviewer believes that answering to this point is necessary to clarify whether cytokine priming induces the production of better EVs from a therapeutic point of view. Unfortunately, the authors are still unable to answer their original question.

Minor comments:

1) Please check the number of references cited in the text and bibliografy: it seems that a mismatch is present

Ref 17 ( Shelke GV, Lässer C, Gho YS, Lötvall J. Importance of exosome depletion protocols to eliminate functional and RNA-containing extracellular vesicles from fetal bovine serum. J Extracell Vesicles. 2014;3(1). DOI:10.3402/jev.v3.24783 ) is not related to the priming of MSCs as cited in the introduction.

Ref 16 (Rodriguez LA, Mohammadipoor A, Alvarado L, Kamucheka RM, Asher AM, Cancio LC, et al. Preconditioning in an Inflammatory Milieu Augments the Immunotherapeutic Function of Mesenchymal Stromal Cells. Cells. 2019;8(5):462. DOI:10.3390/cells8050462) instead should be cited.

2) The first two figures were indicated both as figure 1

3) Figure 3: Histogram showing the expression of CD9 (A) and CD81 (B): A and B are inverted

4) Supplementary figure 2:

- HSP70 shows a different MW between the cells and the EVs. Why?

- Standard MW should be reported next to the first left lane of the WB

- the different experimental conditions should be written on the top of each lane, to make easier to the reader to identify the results

5) It should be considered the possibility of discussing a use of MSC immunoregulatory potential not only in autoimmune diseases, but also in all those pathologies in which the inflammatory component is severe, such as, for instance, in neurodegenerative diseases.

Author Response

We thank this reviewer for their helpful summary and comments on our results. It is always helpful to receive further insight into areas of interest and we feel that addressing the points raised has improved the quality of our analysis.

Comment:  I believe that the study by Hyland and colleagues is well done and the results are convincing. However, it represents only the beginning of the study since MSCs from only 4 donors were used. Therefore, it deserves to be expanded in the future, as pointed out by the authors themselves.

Response:

We thank the reviewer for pointing out the lower n number. Indeed, a larger sample size would be beneficial to this study and we intend for the study to be expanded on in the future. However, we believe that 4 samples represent a valuable and publishable set of data. 

One of the aims of our future work is to find MSC donors that have the most potent/best EVs for treating patients.  We have mentioned the 4 donors as a limitation in terms of EV surface marker expression (Line 295-296), but we have also added a line in the discussion highlighting this limitation and saying that we are working on obtaining a larger sample size to see more differences (Line 364-367) -  “There are limitations in this study which should be considered when interpreting the data. Firstly, our study has a sample size of four and as MSC-EVs are heterogenous, we are working on obtaining a larger sample size to identify further differences between the conditions”.

Despite the low n number, we still feel that our study holds value. At present, there are few published studies on EVs from UC-MSCs and none comparing the specific culture conditions detailed in this study. Additionally, the characterisation of EVs from Quantum derived UC-MSCs represents a further novel aspect of this study as it has not previously been published. Therefore, we think it is worthy of publication given that the limitations have now been further highlighted in the manuscript and hope that the reviewer agrees.

Comment:  The authors showed that MSC stimulation with a cocktail of TNF-a, IFN-g and IL-1b induces an enrichment of pro inflammatory factors in EVs, indicating that the priming of UCMSCs polarize them towards a pro-inflammatory phenotype. The authors well argue that this is in contrast with what has been described in the literature where cytokine priming generally leads to an increase in the immunoregulatory capacity of MSCs and their secretoma, leading to immunosuppression.

In order to clarify this discrepancy, it would be necessary to set up functional experiments in which the EVs extracted from normoxic / hypoxic primed MSCs exert an immunostimulatory action (as the identified cargo would suggest) or immunosuppressant potential (as suggested by most studies in the literature). This could be verified by administrating EVs from the different MSCs on immune cells (e.g. lymphocytes / macrophages or even microglial cells, cell lines or primary cell cultures).

The reviewer believes that answering to this point is necessary to clarify whether cytokine priming induces the production of better EVs from a therapeutic point of view. Unfortunately, the authors are still unable to answer their original question.

Response: We agree with the reviewer here, it is a good suggestion to test the functional effects of the EVs from different conditions. Our lab is aiming to carry out functional experiments on EV phenotypes with lymphocytes in the coming months as part of an ongoing PhD project, but has not been within the scope of this current project.

With regards to it being able to answer the original question, our aim was to clearly characterise and compare populations of EVs grown in different conditions. In order for this to be clear to the reader, we have edited some sentences in the introduction, discussion and conclusion to highlight that this paper presents data on the characterisation of EVs without eluding to their mechanistic properties.

Lines 56-58 (original doc) “This characterisation aims to identify a population of EVs with the best therapeutic potential to supress immune cells in the treatment of autoimmune diseases”.

Has been changed to read:

Lines 57-59 “This paper aims to characterise EV populations from different conditions (normoxia, hypoxia ± proinflammatory conditions) to see if the EV phenotype and cargo varies”.

Line 62-63 (original doc): This information on EV protein cargo is a step towards understanding how UCMSC-EVs maybe capable of suppressing immune cells.

Has been changed to read:

Line 63-64: This information on EV protein cargo is a step towards understanding the biological characteristics of UCMSC-EVs.

Line 362-363 (original doc) ‘Functional studies will be beneficial to see if this EV population alters T-cell polarisation and proliferation’

Has been changed to read:

Line 367-369 (when discussing limitations) ‘Additionally, this study looked solely at the basic characterisation of UCMSC-EVs; and functional studies will be required to see if this EV population alters T-cell polarisation and proliferation, in order to determine their therapeutic potential’

We have changed the first sentence in the conclusion

Line 368-369 (original doc): ‘This research characterised EVs from UCMSCs grown in different conditions with one aim of interpreting how UCMSCs carry out their immunomodulatory effects’

Has been changed to

(Line 374-375) ‘This research represents the first of its kind in characterising EVs from UCMSCs grown in four different culture conditions. It aimed to identify how culture conditions can affect MSC-EV cargo’, so that it is not misleading and clear to the reader.

We believe that we addressed our aim through the series of characterisation experiments detailed in this paper. To our knowledge, this set of comparisons has never been carried out on UC-MSC EVs before and therefore we believe that this research hold value in the UCMSC and EV fields.

Comment: It should also be envisaged to carry out the experiment considering the possibility of comparing EVs extracted from culture media not subjected to freezing.

Response: In our project we froze the EVs after isolation. Freezing EVs in -80oC is a common practice in the EV field as studies have shown that defrosted EVs retain their functional immuno-modulatory effects (Salimu et al., 2017; Di Trapani et al., 2016; Nassar et al., 2016). Because of this, freezing EVs at -80oC is one of the attractive properties of EVs as a potential therapy Additionally, the ultimate clinical implementation of EVs will mandate some form of storage and the preservation of freezing is the likeliest of these. We acknowledge that it would be very interesting to compare EVs stored in -80oC vs freshly prepared to see if this affected the UCMSC EV characteristics, however this was not the aim of this specific paper. But we accept that this may be a very interesting angle to explore in future work.

We hope the reviewers feel our changes have clarified the issues raised and strengthened the understanding and impact of our research.

Salimu J, Webber J, Gurney M, Al-Taei S, Clayton A, Tabi Z. Dominant immunosuppression of dendritic cell function by prostate-cancer-derived exosomes. J Extracell Vesicles. 2017;6(1):1368823.

Di Trapani M, Bassi G, Midolo M, Gatti A, Kamga PT, Cassaro A, Carusone R, Adamo A, Krampera M. Differential and transferable modulatory effects of mesenchymal stromal cell-derived extracellular vesicles on T, B and NK cell functions. Sci Rep. 2016;6:24120.;

Nassar W, El-Ansary M, Sabry D, Mostafa MA, Fayad T, Kotb E, Temraz M, Saad AN, Essa W, Adel H. Umbilical cord mesenchymal stem cells derived extracellular vesicles can safely ameliorate the progression of chronic kidney diseases. Biomater Res. 2016; 20:21. doi: 10.1186/s40824-016-0068-0.

Comment:  Please check the number of references cited in the text and bibliography: it seems that a mismatch is present

Ref 17 ( Shelke GV, Lässer C, Gho YS, Lötvall J. Importance of exosome depletion protocols to eliminate functional and RNA-containing extracellular vesicles from fetal bovine serum. J Extracell Vesicles. 2014;3(1). DOI:10.3402/jev.v3.24783 ) is not related to the priming of MSCs as cited in the introduction.

Ref 16 (Rodriguez LA, Mohammadipoor A, Alvarado L, Kamucheka RM, Asher AM, Cancio LC, et al. Preconditioning in an Inflammatory Milieu Augments the Immunotherapeutic Function of Mesenchymal Stromal Cells. Cells. 2019;8(5):462. DOI:10.3390/cells8050462) instead should be cited.

Response: We put the wrong reference in for that sentence, it has now been corrected. The referencing system has now changed it to no.11 by Rodriguez et al. 2019. (Line 51)

Comment: The first two figures were indicated both as figure 1

Response: Thank you for pointing this out- this has been changed and now labelled correctly (Line 180 and 195).

Comment: Figure 3: Histogram showing the expression of CD9 (A) and CD81 (B): A and B are inverted

Response: Thank you for identifying this - CD9 and CD81 are now correctly labelled (Line 225).

Comment:

4) Supplementary figure 2:

- HSP70 shows a different MW between the cells and the EVs. Why?

- Standard MW should be reported next to the first left lane of the WB

- the different experimental conditions should be written on the top of each lane, to make easier to the reader to identify the results

Response: The difference in MW between cells and EVs is likely due to the presence of multiforms of Hsp70 created by different genes (isoforms) and or post translational modification (i.e. The larger ~75kDa size band present in cells), as well as possibly some non-specific binding to other HSP forms (i.e. The smaller ~65kDa size band present in cells) (Dutta et al., 2013). We know that the antibody used does recognise a range of HSP70 isoforms. The HSP70 detected in the EVs is of the expected molecular weight for HSP70, so we assume that this is positive for HSP70 that is not post translationally modified. It would be interesting to investigate this further, but perhaps beyond the scope of this particular report at this time. 

We have added the standard MWs to the left-hand side and also included a label to each lane that identifies the condition that the sample has come from (Line 39).

Dutta, A., Girotra, M., Merchant, N., Nair, P., & Dutta, S. K. “Evidence of multimeric forms of HSP70 with phosphorylation on serine and tyrosine residues--implications for roles of HSP70 in detection of GI cancers.” Asian Pacific journal of cancer prevention: APJCP vol. 14,10 (2013): 5741-5. doi:10.7314/apjcp.2013.14.10.5741

Comment:  It should be considered the possibility of discussing a use of MSC immunoregulatory potential not only in autoimmune diseases, but also in all those pathologies in which the inflammatory component is severe, such as, for instance, in neurodegenerative diseases.

Response: This is a very good point as the therapeutic scope of MSC-EVs incorporates all conditions with an inflammatory component. We have addressed this comment by rewording autoimmune disease to inflammatory conditions at the start of the Discussion (Line 279). We have also removed the sentence ‘This characterisation aims to identify a population of EVs with the best therapeutic potential to supress immune cells in the treatment of autoimmune diseases’ (Line 57-58 original document) as we felt that it was misdirecting the reader.

Round 2

Reviewer 3 Report

Thank you very much for your professional response.

Major comment 1. To make a long story short, I would be willing to (further) change my recommendation from Reject (or Major Revision now) to Accept if the manuscript is classified as Short Communication not Article (https://www.mdpi.com/journal/cells/instructions).

Additional comments:

Additional comment 1. It would be nice to refer to the recent review published in Cells on mesenchymal stromal cells (https://www.ncbi.nlm.nih.gov/pubmed/31703272).

Additional comment 2. Please, mention that not only proteins are of importance but also miRNAs, bot packed and not packed into the vesicles. This could be very nicely exemplified by asthma (https://www.ncbi.nlm.nih.gov/pubmed/30195377 and https://www.ncbi.nlm.nih.gov/pubmed/31904412).

Reviewer 4 Report

Since the authors declare that functional experiments have already been planned (that was my major concern) and due to the satisfactory answers to my requests, I believe that the study by Dr Hyland and coll. deserves to be published. Over all the revisions done have improved the manuscript.